# Advantages and Disadvantages of Different Treatment Methods in Achondroplasia: A Review

**DOI:** 10.3390/ijms22115573

**Published:** 2021-05-25

**Authors:** Wiktoria Wrobel, Emilia Pach, Iwona Ben-Skowronek

**Affiliations:** Metabolic Laboratory, Department of Paediatric Endocrinology and Diabetology with Endocrine, Medical University in Lublin, Prof. A. Gebala Street 6, 20-093 Lublin, Poland; WiktoriaKW_97@interia.pl (W.W.); emilia.k.pach@gmail.com (E.P.)

**Keywords:** achondroplasia, skeletal dysplasia, short stature, therapeutic drugs, clinical trials

## Abstract

Achondroplasia (ACH) is a disease caused by a missense mutation in the *FGFR3* (fibroblast growth factor receptor 3) gene, which is the most common cause of short stature in humans. The treatment of ACH is necessary and urgent because untreated achondroplasia has many complications, both orthopedic and neurological, which ultimately lead to disability. This review presents the current and potential pharmacological treatments for achondroplasia, highlighting the advantages and disadvantages of all the drugs that have been demonstrated in human and animal studies in different stages of clinical trials. The article includes the potential impacts of drugs on achondroplasia symptoms other than short stature, including their effects on spinal canal stenosis, the narrowing of the foramen magnum and the proportionality of body structure. Addressing these effects could significantly improve the quality of life of patients, possibly reducing the frequency and necessity of hospitalization and painful surgical procedures, which are currently the only therapeutic options used. The criteria for a good drug for achondroplasia are best met by recombinant human growth hormone at present and will potentially be met by vosoritide in the future, while the rest of the drugs are in the early stages of clinical trials.

## 1. Introduction

Achondroplasia is a genetic disorder that belongs to a large group of SDs (skeletal dysplasias) that results in a disproportionate body structure and short stature. The disease is rare and estimated to affect approximately 250,000 people around the world [1,2].

According to the current classification of SDs, achondroplasia is a member of group 1. It is caused by a mutation in the FGFR3 (fibroblast growth factor 3 receptor) gene on chromosome 4p16.3, a mutation that is spontaneous in 80% of cases [3]. The frequency of this mutation notably increases when the father is over 35 years old [4]. The disease is inherited in an autosomal dominant manner and is characterized by full penetration, which means that every patient with a copy of the mutated *FGFR3* gene will exhibit phenotypic characteristics of achondroplasia.

The representative features of achondroplasia have been well recognized around the world for a very long time, even by our ancient ancestors, as reported by [2,5]. The first person to use the phrase ‘achondroplasia’ was Jules Parrot, in 1878, and the description of the disease was initially formulated by Pierre Marie in 1900 and continues to evolve [6,7]. The specific clinical picture of the disease is already apparent in newborns with the condition. In addition to short stature, the clinical manifestations include rhizomelia, short squat trident hands, moderate macrocephaly, dysmorphia and muscle hypotonia. Both the physical features and changes visible in radiographs continue to evolve and become more pronounced with time.

Chronic pain is an integral part of the lives of patients with achondroplasia. The pain is associated with orthopedic and neurological problems, and it increases with the patient’s age. In children, the pain mainly affects the knee joints. In adults, back pain associated with spinal stenosis is predominant, and it often prevents them from working and reduces their economic status [8].

Modern medicine must approach the patient holistically, not only focusing on the treatment of purely physical problems but also seeking to address the effects of achondroplasia symptoms on the patient’s mental health, wellbeing and ability to function in society. This is especially important in the pediatric population because, during this period of life, children who attend school have a natural tendency to compare themselves to others. Unfortunately, such comparisons become increasingly ostentatious and often vulgar, which can lead to a significant decrease in children’s self-esteem and even cause auto-aggressive disorders. Children with achondroplasia have a sense of otherness. This results in isolation from their peer group and, consequently, delayed social development and communication skills. The difficulties related to short stature include affecting everyday functioning, such as problems with participating in physical activities and playing with others, which translate into feelings of irritation, anger and, consequently, loneliness. [9].

In order to understand the difficulties that patients face, a special scale assessing the quality of life with achondroplasia was created, called the Achondroplasia Personal Life Experience Scale (APLES), which includes the perspective of both the young patient and his or her parents [10]. An analysis was also conducted among children between 2 and 12 years of age, called the Achondroplasia Child Experience Measures (ACEMs), which identified both major health problems (ACEM-Symptom) and their negative effects on life (ACEM-Impact) [9]. The development of standardized scales, such as APLES and ACEMs, is very important for developing new treatments and assessing their effectiveness and actual benefits to patients. There have been many publications on the pathogenesis of this disorder and potential therapeutic approaches, but to date, no drug has been developed that positively affects the proportions of the body, rather than just growth.

The aim of this review was to analyze the pharmacological therapeutic methods in achondroplasia, including the advantages and disadvantages of all drugs, both those currently used and those in different phases of clinical trials that have been demonstrated in human and animal studies.

### 1.1. Epidemiology

A comprehensive meta-analysis of the data on the worldwide prevalence of achondroplasia was recently carried out, in which publications from the last half-century were analyzed.

The authors of this review estimated a disease prevalence of 4.6 per 100,000 births, and they noted large regional variation, i.e., higher rates in North Africa and the Middle East than in North America, South America or Europe. However, a limitation of this analysis was the differences in the methodological quality of the compared studies [11].

A meta-analysis containing data from a multicenter study in Europe shows that the achondroplasia prevalence is 3.72 per 100,000 births. The authors analyzed 434 cases, including 350 live births, 82 post-diagnosis pregnancy terminations and 2 intrauterine deaths of the fetus. Among the 208 infants in this study with a known family medical history, a de novo mutation in the *FGFR3* gene was the cause of achondroplasia in 166 cases (79.8%), and family-related prevalence of the disease was confirmed in only 42 (20.2%) cases [4].

Due to the more frequent diagnosis of skeletal dysplasia recently and an increased risk of sudden infant death, it is very important to understand this disease entity.

### 1.2. Pathogenesis

The process of discovering new facts about achondroplasia has not followed the sequence of chapters in medical textbooks. Scientists were first able to recognize and name this disease and learned about additional symptoms, but only with the development of genetic engineering were they then able to identify its main cause.

In 1994, there was a breakthrough. First, Le Merrer et al., Velinov et al. and Francomano et al. determined that the achondroplasia gene was located on 4p16.3, and then, two research groups, those of Shiang et al. and Rousseau, identified mutations in the *FGFR3* gene, which are responsible for the majority of achondroplasia cases [12,13].

In 99% of cases, the mutation is at position 1138 and is a single substitution from G to A (guanine to adenine) or G to C (guanine to cytosine), both of which change arginine to glycine at position G380R in the transmembrane part of the receptor [14]. The result of this mutation is the excessive activation of FGFR3, which, in turn, inhibits the development of cartilage cells while also disrupting the bone growth plate. Proper bone development is preserved due to the harmony between the multiplication and maturation of cells and their apoptosis [15].

The FGFR3 receptor includes a large extracellular component (three immunoglobulin-like domains), which also form an FGF (fibroblast growth factor) binding site), a transmembrane part and two intracellular parts with tyrosine kinase activity [2].

Numerous studies on genetically modified mice have revealed how this receptor works and the role it plays in chondrogenesis and bone formation [16].

The FGFR3 receptor, in its baseline state, is a monomer that inhibits cartilage tissue formation by two mechanisms. Firstly, FGFR3 stops chondrocyte proliferation through STAT1 (signal transducer and activator of transcription 1)/p21 (cyclin-dependent kinase inhibitor 1) signaling. Secondly, FGFR3 prevents chondrocyte hypertrophy and differentiation via the MAPK (mitogen-activated protein kinase) cascade: RAS (a family of membrane small GTPases)—RAF (rapidly accelerated fibrosarcoma, a family of serine/threonine-specific protein kinases)—MEK (mitogen-activated protein kinase)—ERK (extracellular signal-regulated kinase) [17].

As a result of the FGFR3 receptor mutation, processes involved in the inhibition of cartilage formation are intensified. This is called a gain-of-function mutation [18]. This inhibits the proliferation and maturation of the growth plate and reduces longitudinal bone growth [15]. The mechanism is possibly related to altered integrin expression [19].

Moreover, FGFR3 signaling through MAPK–BMP (bone morphogenetic protein) leads to the premature closure of the synchondrosis, which is associated with the stimulation of bone tissue formation. This results in the narrowing of the spinal canal and the foramen magnum and neurological problems. The authors concluded that medical intervention must be performed prior to synchondrosis closure in order to effectively prevent these complications [20].

Karrupaiah et al. showed that the transcription factor SNAIL1 (snail family transcriptional repressor 1) is an essential intermediary in this process. It regulates the intracellular signaling pathways of both STAT1 and MAPK [21]. Furthermore, factors such as FGF9, FGF18 and PTHLH (parathyroid hormone-like hormone) also have an additional feedback effect on FGFR3 [22].

Another pathogenic mechanism that may be of interest for the treatment of achondroplasia is a defect in targeted lysosomal degradation in the ubiquitination process. As a result of the mutation and the disruption of the degradation process, the FGFR3 receptor does not undergo proteolysis and further participates in amplifying the signal [23].

A recent discovery must be noted. Authors from France report that the *FGFR3* gene mutation causing hyperactivity also affects immature osteoblasts in addition to chondrocytes, which had been the main focus prior to these results. The growth of long bones was reduced, but moreover, reduced thickness of cortical tissue and craniofacial modifications related to a membranous defect were identified in adult mice with the mutation. These phenomena were not observed when the mutation affected mature osteoblasts [24]. This is important in the context of the treatment of achondroplasia because, when assessing the potential efficacy of a drug, its possible effects on osteoblasts should also be taken into account, irrespective of the effect on chondrocytes.

### 1.3. Clinical Features

Achondroplasia is related to different clinical conditions: these mainly include musculoskeletal system abnormalities but also involve characteristic facial features, neurological complications, obesity, cardiovascular diseases, etc.

Among the most characteristic changes in the skeleton structure are short stature (mean: 131 ± 5.6 cm for men and 124 ± 5.9 cm for women) and rhizomelia, which is the shortening of the proximal parts of the limbs. The typical facial structure is characterized by macrocephaly with a prominent forehead and midface hypoplasia. Another effect of the *FGFR3* mutation and delayed ossification that affects the facial appearance and wellbeing of patients with achondroplasia is hypoplasia and the abnormal tissue formation of the mandible. This interferes with chewing and causes difficulties in communication. In the vertebral column, excessive lumbar lordosis is prevalent, as are kyphosis of the thoracolumbar connection (common in infants) and spinal stenosis (frequent in adults in segments L1–L4). Other musculoskeletal disorders comprise extension and rotation of the elbow, trident hand deformity, brachydactyly, bandy legs and, often, increased joint mobility [2,25,26].

Developmental delay is one of the clinical symptoms of achondroplasia and includes the later achievement of milestones and problems with motor functions and speaking (in patients with loss of hearing). Intelligence is usually preserved, but in the case of central nervous system disabilities, such as hydrocephalus, a decreased level of intellectual function may be observed [27].

Neurological problems are associated with the narrowing of the foramen magnum, which can lead to hydrocephalus and spinal stenosis-induced nerve compression; surgical intervention may be necessary in both situations. Craniovertebral junction stenosis is common and leads to numerous complications: cervical myelopathy, brainstem compression causing respiratory difficulties (which increases the risk of sudden death) and central sleep apnea [28].

Patients with achondroplasia have a predisposition to obesity, which may lead to cardiovascular diseases, joint overload or apnea. Intriguingly, adipose tissue preferentially accumulates in the abdomen (android type) without leading to the typical metabolic complications; on the contrary, patients with achondroplasia have low levels of cholesterol and triglycerides and normal levels of glucose and insulin [25,29].

The next affected system is respiratory. The chest in achondroplasia has small dimensions, which decreases tidal volume, and the small size of the upper airway can cause obstructive sleep apnea [25,30].

A frequent problem is impaired middle ear function, presumably caused by the abnormal development of the Eustachian tube, which results in hearing loss [31].

Due to obesity and others risk factors, e.g., a lack or low level of physical activity, cardiovascular diseases can arise in patients with short stature and contribute to an increased mortality rate [32,33].

Acanthosis nigricans is a less frequent symptom in patients with achondroplasia, but it still occurs in 10% of cases [34].

### 1.4. Diagnosis

There are no diagnostic criteria for achondroplasia, and recognition is based on clinical and radiological symptoms and genetics.

#### 1.4.1. Prenatal Diagnosis

The prenatal diagnosis of achondroplasia includes non-invasive methods such as ultrasound, CT (computed tomography), MRI (magnetic resonance imaging) and cell-free fetal DNA (deoxyribonucleic acid) testing, and the invasive examination of amniotic fluid.

Routine ultrasound during pregnancy, especially in the third trimester, may suggest achondroplasia if the femur length is below the third percentile of the reference range and the “collar hoop” sign is present (rounded overgrowth of periosteum between the epiphysis and metaphysis, and a wider angle for the metaphyseal–diaphyseal junction) [35,36].

Three-dimensional helical computed tomography is also useful for identifying short stature because it can divulge more specific symptoms, e.g., rhizomelia and spinal canal stenosis in lumbar vertebrae, or allow for the better imaging of the “collar hoop” sign [36].

The use of MRI in the prenatal diagnosis of skeletal dysplasia has also been researched and has been shown to confirm the diagnosis in 82% of cases. During the examination, particular attention is paid to the image of the brain, spinal cord, spine and the volume of the lungs. Fetal MRI may be useful in diagnosing different types of skeletal dysplasia, but this is influenced by the skill of the examiner. Magnetic resonance imaging can be a valuable complement to the ultrasound examination, especially when the ultrasound result is inconclusive [37].

The cell-free fetal DNA testing of the mother’s blood is becoming increasingly popular for the diagnosis of congenital diseases because it eliminates the need for invasive amniocentesis. It is possible to detect achondroplasia by the next-generation sequencing of fetal DNA, but this technique is not as widely available as in aneuploidy detection [38,39].

The gold standard for achondroplasia diagnosis remains the genetic examination of the amniotic fluid and identification of a mutation in *FGFR3*, which is located on the short arm of chromosome 4 [35].

Pregnant women who have a high probability of having a child with achondroplasia because of fetal macrocephaly may consider giving birth by cesarean section [25].

All prenatal study results must be confirmed by the postnatal examination of the child.

#### 1.4.2. Postnatal Diagnosis

The presence of characteristic clinical manifestations in combination with radiological signs is the basis of recognizing achondroplasia in infants, and molecular examination is not necessary.

If there is uncertainty due to ambiguous disease symptoms, then evidence of a heterozygous *FGFR3* gene mutation in a proband becomes necessary. The most common analysis used to confirm achondroplasia is the identification of the two most frequent changes: c.1138G>A and c.1138G>C. The next element of the diagnostic process is a multigene panel, which can reveal other substitutions in the *FGFR3* gene, followed by a differential diagnosis [25].

#### 1.4.3. Differential Diagnosis

Achondroplasia must be differentiated from various other conditions that cause short stature: hypochondroplasia, homozygous achondroplasia, thanatophoric dysplasia, SADDAN (severe achondroplasia with developmental delay and acanthosis nigricans) syndrome and others [2].

Hypochondroplasia:

The symptoms of hypochondroplasia are less pronounced than those of achondroplasia. Hypochondroplasia is characterized by short stature with rhizomelic or mesomelic limb shortening, intellectual disability and seizures, but is not associated with trident-shaped hands and facial abnormalities, which are typical in achondroplasia. Less common are developmental delay and neurological problems caused by spinal stenosis [40].

Homozygous achondroplasia:

Homozygous achondroplasia occurs in children of heterozygous achondroplasia parents, and its clinical features are similar to those of thanatophoric dysplasia [2].

Thanatophoric dysplasia:

Thanatophoric dysplasia is a severe, lethal skeletal dysplasia, and it includes two types: TD1 (thanatophoric dysplasia type 1), with telephone-handset-shaped femurs, and TD2 (thanatophoric dysplasia type 2), with a three-leafed clover cranium structure. This disease entity is characterized by growth retardation, macrocephaly with the widening of brain ventricles, frontal bossing and midface retrusion, exophthalmos, brachydactyly, trident-shaped hands and general hypotonia [41].

SADDAN syndrome:

SADDAN (severe achondroplasia with developmental delay and acanthosis nigricans) syndrome is caused by a lysine-to-methionine substitution at nucleotide 650 in the *FGFR3* gene and is related to retarded development, the hyperpigmentation and hyperkeratosis of the skin (called acanthosis nigricans), anatomical abnormalities of the brain, epilepsy attacks and hearing loss [42].

Mutation in the *FGFR3* gene may also cause Crouzon syndrome with acanthosis nigricans, Muenke syndrome, familial acanthosis nigricans and CATSHL (camptodactyly, tall stature, scoliosis and hearing loss) syndrome [43,44,45,46].

Other diseases requiring differentiation from achondroplasia:

Other skeletomuscular diseases have to be distinguished from achondroplasia: cartilage-hair hypoplasia–anauxetic dysplasia, Schmid metaphyseal chondrodysplasia, metaphyseal anadysplasia type 1 and type 2, etc. Different clinical and radiographic symptoms present in these disorders can be used to establish the diagnosis [47].

Despite the nomenclature, pseudoachondroplasia is not difficult to distinguish from achondroplasia, because facial abnormalities that can be observed in achondroplasic infants are absent in children with pseudoachondroplasia. Children with pseudoachondroplasia have normal body lengths after birth, and the growth rate starts to decrease at the age of two years [48].

## 2. Methods of Treatment

The methods for treating achondroplasia can be grouped into surgical and pharmacological therapies. Surgical intervention consists of lengthening the lower limbs using an Ilizarov apparatus or monolateral external fixator and entails multiple procedures and possible serious complications [49].

Due to a better understanding of the pathogenesis of this disease, there have been attempts to develop causal pharmacological therapies.

### 2.1. C-Type Natriuretic Peptide in Achondroplasia Treatment

CNP (C-type natriuretic peptide) plays an important role in human longitudinal growth. It was first described in 1998 when the expression of CNP mRNA was detected in the tibia of fetal mice, suggesting that the CNP pathway played a role in the ossification process [50]. During this process, chondrocytes first proliferate, followed by hypertrophy, apoptosis and the appearance of osteoclasts, bone marrow cells and, finally, osteoblasts, which produce bone, consequently changing cartilage into bone tissue. Endochondral ossification is also regulated by many systemic and local factors, e.g., GH (growth hormone), thyroid hormone, IGF-1 and IGF-2 (insulin-like growth factor 1 and 2), Ihh (Indian hedgehog), PTH (parathyroid hormone) and fibroblast growth factors [51].

Natriuretic peptides can be differentiated into A, B, C and D types, which activate three subtypes of natriuretic peptide receptors: NPR-A, NPR-B and NPR-C [52]. ANP (atrial natriuretic peptide) and BNP (brain natriuretic peptide) regulate the functions of the circulatory and excretory systems, have diuretic and natriuretic effects, reduce the fibrosis and remodeling of the heart muscle, relax smooth muscles in the vascular walls and cause vasodilatation. CNP has less of an effect on the cardiovascular system and renal excretion, and its main effect is the stimulation of chondrocytes and the growth of long bones. The last of the natriuretic peptides is DNP (dendroaspis natriuretic peptide), which appears to have vasodilatory properties in animals, successfully competes with ANP for receptor binding and inhibits the activity of L-type calcium channels in the heart, but its exact function is not yet known [53,54].

The C-type natriuretic peptide acts on the growth plate through the NPR-B receptor, causing the transformation of GTP (guanosine 5′-triphosphate) into cGMP (cyclic guanosine monophosphate) and the activation of mediators, including phosphodiesterases, and cGKI and cGKII (cGMP-dependent protein kinases I and II), mainly cGKII. The kinases inhibit the MAPK pathway by inhibiting RAF-1 and, consequently, MEK-1, MEK-2, ERK-1 and ERK-2. This leads to gene expression in the chondrocyte nucleus, stimulates the proliferation and differentiation of chondrocytes and increases the formation of the extracellular matrix [52].

The overexpression of CNP has been shown to increase endochondral ossification in chondrocytes in transgenic mice. Increased body lengths were observed in transgenic mice compared to the rest of the litter, including elongation of the femur, spine and skull, which can widen the foramen magnum and reduce the need for surgical intervention. A decrease in the mineral density of the femur and spine was noted in these animals [55].

Natural C-type natriuretic peptide has a short half-life of 2–3 min, as it is rapidly degraded by NEP (neutral endopeptidase). For this reason, its effective use in the treatment of achondroplasia is almost impossible, and attempts have been made to use other forms of this substance [56]. Two independent studies on the use of CNP in achondroplasia therapy were recently performed: the first is a vosoritide (BMN-111) trial, and the second is a TransCon CNP study, which is currently underway (see Table 1, Table 2 and Table 3).

#### 2.1.1. C-Type Natriuretic Peptide Analog: Vosoritide (BMN-111)

Vosoritide is resistant to natural neuroendopeptidase, as shown by clinical studies, and is a promising treatment for achondroplasia. A small molecule that acts on the NPR-B receptor is able to diffuse freely into the growth plate. In studies carried out on transgenic mice and cynomolgus monkeys, an increase in the growth of long bones and the tail was observed, and consequently, the growth of the axial and appendicular skeleton increased. Monkeys treated with BMN-111 for six months showed widening of the tibial growth plates with an increase in the hypertrophic zone, as well as dilation of the lumbar vertebral openings. At the same time, only transient, mild hemodynamic changes (a decrease in blood pressure and increase in heart rate) were observed in the test animals [57].

The third phase of the vosoritide clinical trial consisted of 121 patients aged 5–18 years who were randomly assigned to groups (61 taking the placebo and 60 taking vosoritide), in which 15 µg/kg body mass vosoritide or placebo was administered in once-daily subcutaneous injections for 52 weeks. The results of this trial are very promising. The least-squares mean growth velocity in children taking vosoritide was 1.71 cm/year, in comparison with a gain of 0.13 cm/year in patients receiving placebo, with a difference of 1.57 cm/year in favor of vosoritide. However, the higher velocity is lower than that in healthy prepubertal children (mean, 5 cm/year). No serious side effects related to drug administration were observed, and mild effects were reactions at the injection site and a slight, temporary drop in blood pressure. Antibodies to vosoritide developed in 42% of patients, but they did not neutralize the drug and did not lead to an increase in the hypersensitivity reaction. Treatment with vosoritide did not result in abnormal bone maturation or problems with the proportionality of the upper and lower body parts of the patients [58].

In 2020, a clinical trial was launched to test the effectiveness of vosoritide in reducing foramen magnum stenosis (NCT0455494) and compression of the spinal cord at the craniocervical junction site, therefore potentially reducing the need for surgical decompression [59] (see Table 1, Table 2 and Table 3).

#### 2.1.2. Prodrug, Prolonged-Release C-Type Natriuretic Peptide: TransCon CNP

Unlike vosoritide (CNP-39), which consists of 37 amino acids from natural CNP and glycine and proline at the N-terminus of the polypeptide, TransCon (CNP-38) contains 38 amino acids (37 from natural CNP and lysine at the N-terminus of the polypeptide). These changes in the structure of both molecules and the fact that TransCon CNP is more resistant to the action of NEP mean that their half-lives are significantly different, amounting to about 15–20 min for vosoritide and 90 h for TransCon, while in blood, vosoritide can be detected up to about 2 h after administration, and TransCon is detectable for up to 7 days. TransCon’s mechanism of action and its effects on body tissues, including the growth plate, are similar to those of vosoritide. The long-release form of the drug will help to avoid high blood CNP peaks and associated adverse cardiovascular effects [56].

A study of the effectiveness of CNP-38 in monkeys showed that, after the administration of TransCon CNP at a dose of 100 µg/kg per week, the increase in height in the test group was 5% greater than that in the control group. With for the daily administration of CNP-39, the height increase was 3% in the test group, and the tail length increased successively by 9% and 3% compared to the control. No adverse effects on bone quality were observed in the tested animals. Studies have also shown an increase in the width of the proliferative zones in the proximal tibia in the group receiving the drug. The administration of CNP-38 at 203 µg/kg per day via subcutaneous injections or continuous infusion in mice resulted in significant growth of the axis and limb skeleton. The effect was more pronounced with continuous infusion; in comparison with the control group, the treatment resulted in increases of 7.1% in femoral length, 12.2% in tibia length and 25% in spinal length, while daily subcutaneous administration induced increases of 5.5%, 4% and 11.3%, respectively. The adverse events in mice were a dose-dependent decrease in blood pressure without the heart rate being affected, while there were no significant differences in monkeys [56].

Phase 2 (which started in June 2020) of the TransCon CNP clinical trial is ongoing. The drug is administered subcutaneously once per week, which will continue for one year, in children with achondroplasia aged 2–10 years (NCT04085523) (see Table 1, Table 2 and Table 3).

### 2.2. Recombinant Human Growth Hormone (rhGH)

Growth hormone (somatotropin) is an anabolic hormone involved in the synthesis of nucleic acids and proteins, the stimulation of cell division and the regulation of carbohydrate metabolism, which, in effect, causes the growth of organs and long bones, as well as weight gain [60].

Recombinant somatotropin is one of the symptomatic treatment methods for short stature in achondroplasia and aims to enhance the growth of the patients via direct action or through the effect of IGF-1 on chondrocyte proliferation [61].

Several studies have confirmed that short-term GH therapy is more effective than long-term treatment in increasing growth velocity. Notably, the greatest height gain may be observed in the first year of taking the medication [62,63].

Human recombinant growth hormone (rhGH) treatment for about 10 years resulted in a mean growth gain of +3.5 cm in men and +2.8 cm in women. The combination of rhGH and l-thyroxine resulted in a final growth increase of +10.0 cm in males and +9.8 cm in females. Combining this method with surgical tibial and/or femoral elongation increased the final height by +17.2 cm and +17.3 cm in males and females, respectively [64].

Growth hormone treatment has been found to be ineffective in patients with deformities of the lower limbs and spine [62].

Studies in rats did not show a significant difference in height gain between the growth hormone and placebo groups, but only the treatment group showed greater weight gain. The authors also noted that the use of a variable GH treatment model (more similar to the natural secretion rhythm) may be more effective than continuous daily administration, but this requires further examination [65].

To date, it has not been confirmed whether the administration of somatropin negatively affects the severity of foramen narrowing and pressure on the spinal cord, and no symptoms of acromegaly have been observed in the treated patients [63].

A meta-analysis of recombinant human growth hormone treatment in achondroplasia based on an extensive group of patients (n = 558) shows that data about body disproportion in GH treatment are ambiguous [63] (see Table 1, Table 2 and Table 3).

### 2.3. Tyrosine Kinase Inhibitor (TKI): Infigratinib

Infigratinib (NVP-BGJ398/BGJ398) is an orally administered tyrosine kinase inhibitor of the FGFR receptor family. It also shows anti-angiogenic activity and therapeutic potential in the field of oncology: recently, it was pre-registered in Australia/Canada for the treatment of cholangiocarcinoma [66].

In 2016, selected tyrosine kinase inhibitors were compared in terms of selectivity for the FGFR3 receptor as well as their possible use in the treatment of achondroplasia, and the results indicated that NVP-BGJ398 was the best candidate. However, because its specificity is too low and its toxicity to other organs is too high, it was deemed to be a poor therapeutic strategy, and it was suggested that other options be explored [67].

In the same year, French authors came to completely different conclusions after conducting studies on human and mouse chondrocytes and mouse models of achondroplasia [68].

Their results cast new light on the NVP-BGJ398 inhibitor. They confirmed the inhibition of FGFR3 phosphorylation and its decreased expression in chondrocytes and concurrently found no effect on FGFR1, the level of FGF23 or blood phosphorus levels, which again proves the high selectivity of this inhibitor. The use of the inhibitor returns FGFR3 to its normal level of activity. In molecular terms, it reduces phosphorylation in a RAS-ERK-dependent manner and reduces the transcription of SOX9 (SRY-box transcription factor 9) and STAT1 [68].

In the tested ACH mouse models, treatment with the NVP-BGJ398 inhibitor caused elongation of the femur, alleviated the disruption of chondrocyte differentiation and increased the growth of all four limbs (range between +11.9% and +32.6%) and the tails of mice. Moreover, correction of the size of the L4–L6 (lumbar vertebra 4–6) sections of the spine and beneficial effects on the intervertebral discs were evident, which may potentially positively affect the risk of spinal stenosis and postural defects accompanying achondroplasia [68].

Another advantage that may encourage further research is the improvement of mandibular anomalies, which also stems from the defective differentiation of cartilage tissue. By improving the shape of the lower jaw, the use of the NVP-BGJ398 inhibitor may indirectly alleviate the defect in the respiratory system, which causes obstructive sleep apnea [26].

The important findings are the increase in the size of the foramen magnum and the retention of the premature fusion of the synchondroses [68].

The authors that carried out this experiment compared their results to BMN111, and the NVP-BGJ398 inhibitor was more effective in ameliorating the dwarf phenotype in the limbs, tail and spine [68].

Unfortunately, NVP-BGJ398 did not appear to improve the defect in the structure of long bones. The authors attributed this shortcoming to the treatment period being too short [68].

Dermatological problems, such as dry skin and mucous membranes, nail plate disorder, alopecia and hand–foot syndrome, have been reported during the use of infigratinib in oncology, but we do not know if such effects will occur when using doses of this drug for the treatment of ACH [69].

All optimistic results obtained in murine models led to the initiation of a clinical trial on the possibility of using infigratinib for the treatment of achondroplasia in humans.

Phase II of the PROPEL trial (NCT04265651) by QED Therapeutics is currently underway to provide information on the efficacy and possible side effects of infigratinib, but the results will not be known for some time because the estimated study completion date is July 2026 (see Table 1, Table 2 and Table 3).

### 2.4. Soluble FGFR3 (TA-46): Recifercept

Recifercept (previously TA-46) is an artificially produced sFGFR3 (soluble FGFR3) that competitively binds to members of the FGF family (FGF2, FGF9 and FGF18 are confirmed), thus limiting the activation of the defective receptor in achondroplasia [70].

Recifercept exerts its effects through the downregulation of the MAPK and phospholipase C pathways (only in an FGF-dependent manner) but not via STAT1 [70].

The results of preclinical studies in mice show a number of benefits of this drug, such as the restoration of bone growth, improvement in structure, the elongation of long bones, improvement of the skeletal development of the thorax, an increase in cortical bone thickness and even a reduction in mortality. There were no adverse effects or loss of ability to procreate [71].

Moreover, better parameters of the axial skeleton were achieved, including the alleviation of kyphosis and side effects of lumbar compression and the restoration of proper vertebral maturation. Treatment corrected the proportions of the skull by preventing premature closure of the synchondroses [71].

Interestingly, tests on the reproductive capacity after recifercept therapy confirmed the enlargement of the pelvic bone size. Taking this and the improved proportion of the skull into account, recifercept could reduce the need for cesarean sections in women with achondroplasia [71].

There are reports of an advantageous effect of recifercept (if it is implemented early enough) on atypical visceral obesity in achondroplasia. In animal models, Fgfr3 mutations have been associated with an elevated predisposition to adipogenesis in mesenchymal stem cells. In mice treated with sFGFR3, this phenomenon was eliminated [29] (see Table 1, Table 2 and Table 3).

### 2.5. Vofatamab (B-701)

Vofatamab is a human monoclonal antibody that binds to the external domain of FGFR3, thus preventing the attachment of ligands to the FGF family [72].

This drug is currently under investigation in clinical trials for use in urothelial neoplasms and multiple myeloma [73]. Recently, it was also considered as a potential therapeutic option in achondroplasia [74]. However, no preclinical studies have been initiated to date.

It should be noted that a possible problem is that the size of the antibody is too large. It is difficult to predict whether it will penetrate the growth plate, where the extracellular matrix is fairly dense (see Table 1, Table 2 and Table 3).

### 2.6. Meclizine

Meclizine is an antihistamine drug commonly used to relieve the symptoms of travel sickness or dizziness. Its additional mechanism of action is blocking the ERK1/2–MAPK pathway, which inhibits the activity of FGFR3 [75].

In preclinical studies, similar to previously described drugs, meclizine increased the length of the body and the bones of the skull, fore and hind limbs, and L1–L5 vertebrae. Surprisingly, these effects were also observed in mice without the ACH mutation. Additionally, it strengthened the trabecular bone by increasing its thickness [76,77].

However, it was unsuccessful in affecting the size of the foramen magnum or lumbar spinal stenosis [76].

Despite the fact that meclizine is easily available and often used, in the treatment of achondroplasia, it is necessary to administer it repeatedly and, importantly, during the growth phase in children.

The results of a Phase Ia study on the safety and appropriate dose of meclizine in children with achondroplasia were recently published. The drug was well tolerated, and there was no evidence of toxic accumulation resulting in visible side effects [78].

Further information will be obtained in Phase II, the beginning of which is not yet planned (see Table 1, Table 2 and Table 3).

### 2.7. Statins

Other widely used drugs are statins, which are inhibitors of HMG-CoA (3-hydroxy-3-methylglutaryl-coenzyme A) reductase. Because of their anti-atherosclerotic effects, they are used in the treatment of hypercholesterolemia [79].

In 2015, with the use of iPScs (induced pluripotent stem cells) created from the epidermis of people with achondroplasia, it was suggested that statins could induce the proteasomal degradation of mutant FGFR3. The drugs improved the formation of cartilage tissue, increased the proliferation of primary cartilage and increased the growth of long bones [80].

Unfortunately, three years later, completely different conclusions were drawn. An effort to explain the exact mechanism of action revealed that statins have no effect on FGFR signaling in chondrocytes. The article, however, did not completely rule out statins. Further research is needed to clarify whether statins actually have a positive effect on chondrocytes and, if so, the underlying mechanism [81] (see Table 1, Table 2 and Table 3).

### 2.8. Parathyroid Hormone (PTH) and Parathyroid Hormone-Related Peptide (PTHrP)

Parathyroid hormone is involved in endochondral ossification and affects the proper development of cartilage tissue during the growth process as early as in the womb.

Parathyroid hormone and parathyroid hormone-related peptide act on growth plate cells through a common PPR receptor (PTH/PTHrP receptor) and cause the proliferation and differentiation of chondrocytes, increase the amount of extracellular matrix and stimulate the differentiation of mesenchymal cells into limbs [82,83,84].

Animal studies indicate that PTH has a positive effect on growth velocity, and body lengths were similar between PTH-treated mice with achondroplasia and the rest of the healthy litter. PTH is also beneficial for the cranial shape, presumably due to the retardation of the premature fusion of skull synchondroses. A very interesting finding is that PTH inhibits the activation of *FGFR3*, the gene that causes achondroplasia. Parathyroid hormone-related peptide mRNA expression in chondrocytes is greater because of PTH action, and PTHrP promotes the proliferation of these cells, which contributes to improving linear growth [84].

The long-term effects of PTH treatment are unknown, and further studies on the safety profile are required, but recombinant human PTH may be a potential treatment option for achondroplasia (see Table 1, Table 2 and Table 3).

### 2.9. FGFR Inhibitor: ASP5878

ASP5878 is a new inhibitor of FGFR1-4 receptors. Its effectiveness in treating hepatocellular and urothelial tumors, including those caused by FGFR3 defects, suggests a possible benefit in the treatment of achondroplasia [85].

The inhibitor effect was analyzed and determined to be less effective for targeting bone elongation than the CNP analog. However, attention was drawn to the possibility of oral ASP5878 administration, which may reduce the stress and inconvenience associated with multiple subcutaneous injections of the CNP analog in pediatric patients suffering from achondroplasia [86].

The use of ASP5878 increased the thickness of the growth plate. Moreover, when examining iPSC cells, an effect on chondrocyte equivalents was found [86].

The authors of the publication recommend caution due to the emerging atrophy of the corneal epithelium in mice and the increased frequency of quite numerous side effects during the use of the drug in adult cancer patients [86].

Other molecules considered for use as FGFR3 inhibitors in ACH are NF449, A31 and P3. For NF449 and A31, the target is tyrosine kinase blockade, while P3 binds the extracellular domain of the receptor. Currently, no data are available as to whether further studies are being carried out on their use [75] (see Table 1, Table 2 and Table 3).

### 2.10. Aptamer RBM-007

Aptamers are single-stranded molecules of DNA or RNA (ribonucleic acid) and are formed by a method called SELEX (systematic evolution of ligands by exponential enrichment) [87].

The mechanism of action of an aptamer is very similar to that of antibodies: it can bind to targets with high affinity and specificity. However, aptamers have the advantages of smaller size, lower antigenicity and lower production costs.

In the context of the treatment of achondroplasia, an aptamer called RBM-007 is presently being tested. RBM-007 is an anti-FGF2 aptamer and specifically binds to FGF2, preventing its binding to the FGFR3 receptor. Previous research publications on mouse models have drawn optimistic conclusions regarding the use of aptamers in diseases related to the skeletal system [88].

Authors reported that RBM-007 rescued the impaired differentiation and maturation of chondrocytes and restored defective skeletal growth in a mouse model of achondroplasia [89,90]. RBM-007 is in a Phase 1 study to evaluate its safety, tolerability and pharmacokinetics (JapicCTI-205345).

The mechanisms of the currently used and experimental methods of pharmacological treatments for achondroplasia are shown in Figure 1.

All of the aforementioned potential therapeutic medications that are currently being explored have advantages as well as disadvantages, which are summarized in Table 2 and Table 3. There is a risk that a cure for all ailments will not be found. In that case, perhaps a therapy that involves the combined use of several different drugs should be tested. There are already precedents confirming the efficacy of such a procedure. A great example is the use of the phosphatase inhibitor LB-100 in combination with BMN-111. Their synergistic effects in increasing growth in bone length have been demonstrated [91].

## 3. Conclusions

All of the above-described data demonstrate that finding a cure for achondroplasia is the subject of very intensive research by teams across the globe. At present, investigations into rhGH and vosoritide are the most advanced. However, even if these treatments are cleared for widespread use, they will not cure all the symptoms associated with achondroplasia. Although they provide the benefit of increasing bone length, their effects on important aspects, such as disproportionality, the axial skeleton and the foramen magnum, have unfortunately not been confirmed. Each of these aspects entails further complications that leave their mark on the daily lives of patients with achondroplasia.

The ideal drug for achondroplasia should be small in size, to readily penetrate the growth plate, be specific for FGFR3 and effectively inhibit its signaling pathway. As the therapy is long term, its production costs should be as low as possible, and the form of drug administration should be easy and acceptable for a pediatric patient. In addition, side effects should be minimized to the level of dose tolerance.

The criteria for a good drug for achondroplasia are best met by recombinant human growth hormone at present and will potentially be met by vosoritide in the future, while the rest of the drugs are in the early stages of clinical trials.

## Figures and Tables

**Figure 1 ijms-22-05573-f001:**
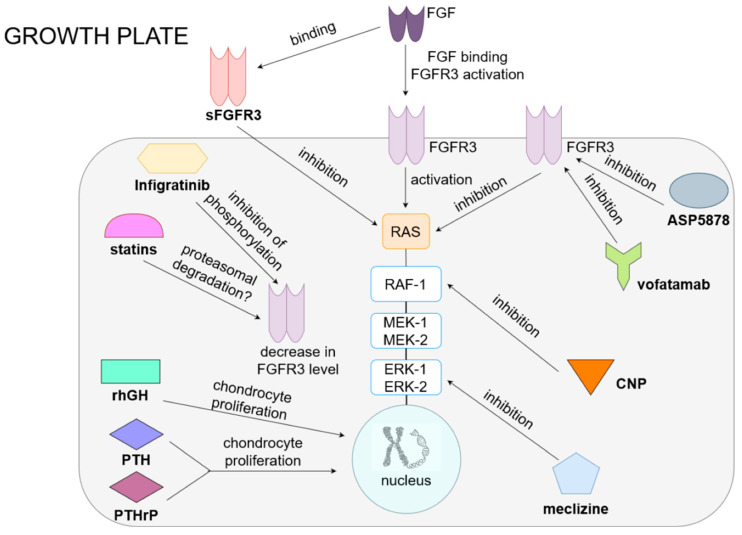
Current and potential treatments for achondroplasia. In the normal growth plate, FGF acts on FGFR3, causing the activation of RAS and the MAPK pathway (RAF, ERK and MEK), resulting in the inhibition of chondrocyte proliferation and differentiation and a reduction in extracellular matrix synthesis. Drugs for achondroplasia include preparations that influence the function of the FGFR3 receptor and MAPK pathways and directly act on chondrocytes. The first group includes, among others, drugs that inhibit the receptor (ASP5878, which is an inhibitor of FGFR3, and vofatamab, an anti-FGFR3 antibody), as well as drugs that reduce the concentration of FGFR3 (infigratinib, which inhibits receptor phosphorylation, or statins, potentially causing proteasomal degradation of this molecule). The drug that competes with FGFR3 for a ligand is sFGFR3, a soluble receptor that, by binding to FGF, reduces its effect on the growth plate. Drugs that inhibit the MAPK pathway are CNP analogs, which inhibits RAF-1, and meclizine, which inhibits ERK. By acting on FGFR3 or MAPK, all of these drugs consequently increase the proliferation and differentiation of chondrocytes, improving bone elongation growth. RhGH and PTH or PTHrP have a direct stimulating effect on the proliferation and differentiation of chondrocytes.

**Table 1 ijms-22-05573-t001:** Classification of drugs according to the target of action.

Target of Action	Drug Name
FGFR3	sFGFR3 (recifercept)
	Inhibitors of HMG-CoA (statins)
	Tyrosine kinase inhibitor (infigratinib)
	FGFR 1–4 inhibitor (ASP5878)
	FGFR3 antibody (vofatamab)
NPR-B	CNP analogue (vosoritide)
	Sustained-release CNP prodrug (TransCon CNP)
MAPK pathway	Meclizine
Chondrocyte nucleus	rhGH
	PTH/PTHrP

Abbreviations: CNP—C-type natriuretic peptide; FGFR—fibroblast growth factor receptor; HMG-CoA—3-hydroxy-3-methylglutaryl-coenzyme A reductase; NPR-B—natriuretic peptide receptor B; MAPK—mitogen-activated protein kinase; PTH—parathyroid hormone; PTHrP—parathyroid hormone-related peptide; rhGH—recombinant human growth hormone; sFGFR3—soluble fibroblast growth factor 3 receptor.

**Table 2 ijms-22-05573-t002:** Benefits of different drug therapies in achondroplasia.

Drug Name	Benefits for People	Benefits for Animals
	*Clinically used drugs*	
Recombinant human growth hormone (rhGH)	Possibly better growth pattern in children with achondroplasia, especially in combination with l-thyroxine and surgical elongation of tibia and/or femur	Possibly better growth velocity with variable rather than continuous drug administration
	*Drugs in different phases of clinical trials*	
C-type natriuretic peptide (CNP) analog: vosoritide	Treatment targets underlying molecular pathogenesis, increasing growth velocity and height Z-score More proportional growth Resistance to natural endopeptidase No serious side effects	Increase in axial and appendicular skeleton growth Increase in the hypertrophic zone in tibial growth plates Widening of lumbar vertebral openings
C-type natriuretic peptide prolonged-released: TransCon CNP	Long half-life, about 90 h Resistance to natural endopeptidase Prevents adverse cardiovascular effects because of long-release form Phase II of research in progress NCT04085523	Increase in body and tail length in monkeys No adverse effect on bone quality Increase in the width of the proliferative zones in the proximal tibia
Infigratinib (NVP-BGJ398)	No data Phase II of research in progress NCT04265651	Increases the growth of long bones, axial and craniofacial skeleton Increases the size of foramen magnum Correction of spinal stenosis Ameliorates the defective differentiation of the chondrocyte
Soluble recombinant human fibroblast growth factor receptor 3 (soluble FGFR3): recifercept	No data Phase II of research in progress NCT04638153	Reduces mortality Restores skeletal bone growth Increases cortical bone thickness Decreases spinal and skull deformities Enlargement of pelvic bone Corrects metabolic alteration (helps with atypical obesity)
Vofatamab (monoclonal antibody specific for FGFR3)	No data	No data
Meclizine	Administered orally No data	Increases the growth of long bones, axial and skull lengths Ameliorates short stature Increases trabecular thickness
Statin	Ambiguous data	Ambiguous data
ASP5878	Administered orally	Increases the growth of long bones Elongates the length of the cranial base Increases thickness of growth plate cartilage
Parathyroid hormone (PTH)	Causes proper development of cartilage tissue Increases proliferation and differentiation of chondrocytes and mesenchymal cells, extracellular matrix synthesis	Positive effect on growth velocity, similar body length to the rest of the healthy litter Retardation of premature fusion of the skull synchondrosis Inhibition of FGFR3 activation

**Table 3 ijms-22-05573-t003:** Drawbacks of different drug therapies in achondroplasia.

Drug Name	Drawbacks for People	Drawbacks for Animals
	*Clinically used drugs*	
Recombinant human growth hormone (rhGH)	Theoretical possibility of the appearance of acromegaly signs, increase in foramen magnum narrowing and spinal cord compression, but no conclusive evidence Ineffective in the case of deformation of the limbs and spine Requires daily subcutaneous injections	Increases body mass
	*Drugs in different phases of clinical trials*	
C-type natriuretic peptide (CNP) analogue: vosoritide	Mild side effects: transient changes in blood pressure Requires daily subcutaneous injections	Transient, mild hemodynamic effects
C-type natriuretic peptide prolonged-released: TransCon CNP	No data Phase II of research in progress NCT04085523	Dose-dependent lowering of blood pressure in mice but not in monkeys
Infigratinib (NVP-BGJ398)	Administered in injections No data Phase II of research in progress NCT04265651	No effect on the defect in the structure of long bones Not found
Soluble recombinant human fibroblast growth factor receptor 3 (soluble FGFR3): recifercept	Administered in injections No data Phase II of research in progress NCT04638153	No effect on the trabecular bone Effects are mediated only through FGF-dependent pathway No signs of toxicity Preserved fertility
Vofatamab (monoclonal antibody specific for FGFR3)	No data	No data
Meclizine	Phase I completed: no signs of toxicity were found	No effect on the area of foramen magnum or lumbar spinal canal Cumulative effect of 20 mg/kg—toxicity: ineffective bone growth
Statin	Ambiguous data	Ambiguous data
ASP5878	Hyperphosphatemia Retinal detachment Diarrhea Elevated alanine transaminase	Slight atrophy of the corneal epithelium
Parathyroid hormone (PTH)	Unknown long-term effects, need for further studies on the safety profile	No data

## Data Availability

Not applicable.

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
