# Peer review of "Advantages and Disadvantages of Different Treatment Methods in Achondroplasia: A Review"

_ijms, 2021, doi:10.3390/ijms22115573_

Round 1

Reviewer 1 Report

The paper of Wrobel and coworkers was aimed to reviewing the different treatment methods in achondroplasia. The authors described the potential and available therapeutic options taking into account the advantages and disadvantages of all drugs. The subject matter is of interest, however the paper is redundant and poorly focused on the main topic.

Specific comments.

  • The sections devoted to epidemiological, pathogenesis and clinical of achondroplasia should be reorganized and summarized as they did not represent the main focus of the study. In this regard recent exhaustive reviews have been published (Hogler and Ward, Wien Med Wochenschr 2020; 170 (5) :104-111; Pauli, Orphanet J. Rare Dis. 2019; 14:1)
  • The section on drugs should be reorganized providing for a better readability a classification based on their pharmacodynamics (e.g. Drugs targeting the FGFR3 and downstream signaling, Drugs targeting the CNP receptor NPR-B and so on)
  • The drugs targeting the FGFR3 ligands (i.e. Fibroblast growth factor 2 aptamer (RBM-007) and Soluble FGFR3 decoy (TA-46)) should be added
  • The pharmacokinetic data, where available should also be included
  • English language should be carefully revised

Author Response

Response to Reviewer 1

Thank you to for your valuable advice and remarks. We have decided to make the changes according to your suggestions.

The sections devoted to the epidemiological, pathogenesis, and clinical aspects of achondroplasia have been minimally reorganized, as we would like our review to be useful to clinicians.

We have cited the publications mentioned: Hogler and Ward, Wien Med Wochenschr 2020; 170 (5) :104-111; Pauli, Orphanet J. Rare Dis. 2019; 14:1.

According to your suggestions, the section on drugs is started by Table 1 with classification of drugs based on pharmacodynamics for better readability.

Soluble FGFR3 was formerly called TA-46, but in our review, we use the name Recifercept. In accordance with your valuable comments, we have added a section on RBM-007.

The pharmacokinetic data were unavailable in a complete form. In our opinion, it is impossible to compare the pharmacokinetics of drugs before the publication of all results of the trials.

The English language has been carefully revised and  has been send to English pre-edit services of MDPI.

Reviewer 2 Report

The review entitled “Advantages and disadvantages of different treatment method in achondroplasia – review” is an interesting review. However, I have some comments to improve it.

Line 56: the full stop should be deleted.

At the end of the introduction, the authors should specify the aim of the present review.

Lines 83-85: the sentences should be rewritten.

The references should be reported after the full stop.

Lines 90-91: “Multicenter studies in Europe shows that achondroplasia prevalence is 3,72 per 100000 birth.” The authors cite these “multicenter studies”.

Lines 91-94: “434 cases were analyzed, including 350 live births, 82 post-diagnosis pregnancy terminations and 2 intrauterine deaths of the fetus. De novo mutation in FGFR3 gene was cause of achondroplasia in 166 from 208 infants (79,8%) with known family medical history, only in 42 (20,2%) cases confirmed family-related prevalence of the dis-94 ease. [4]”  This part is not clear. In the first sentence, the authors discuss about 434 cases, while in the second the authors report a total of 208 infants. Did the authors refer to two different studies?

line 108: Truncated forms should be not used.

Line 111: “ transformation” should be changed to “mutation”.

The authors could add a figure about FGFR3 receptor. Moreover, a figure reporting the physiological function and the gain of function occurring as a result of the mutation could be useful.  

Line 144: “(Zhang et al., 2006).” should be corrected.

Line  142-143: the authors reported that the expression is low. It is not clear what kind of expression is low (protein or gene) and if they referred to FGFR3.

Line 272: the full stop should be deleted.

Lines 366-384: It seems that the authors described two studies: one on mice and the other on monkeys. However, the authors provided only one reference.

Line 462: the authors reported that several clinical studies started but they described only one.

Lines 563-564 and lines 568-570: the sentence should be rewritten.

The section 2 regarding the methods of treatment should be improved. The authors reported in the same section promising molecules that still in a preclinical phase and also molecules at a clinical phase. This is confusing for the reader. I suggest to report molecules under investigation in clinical trials in section 2 and add a section 3 reporting molecules in a preclinical stage. Moreover, the authors should clearly report those molecules that are currently used to treat the pathology in the first part of section 2.  The tables should be modified accordingly and references should be added.

Abbreviations should be defined and used consistently throughout the manuscript. For example: line 236, “fibroblast growth factor receptor 3 gene” should be abbreviated.  Line 299: “CNP” should be defined.

The authors should check the manuscript in order to use decimal point and not decimal comma. For example: lines 90,93, 94 and 171.

Author Response

Response to Reviewer 2

Thank you to for your valuable advice and remarks. We have decided to make the changes according to your suggestions.

The article has been corrected in terms of punctuation; in line 56 and in line 272 (285 in the new version), the full stop has been removed.

As suggested, the purpose of the review (lines 82-85) has been added at the end of the introduction.

The text in lines 83-85 (90-92 in the new version) has been redrafted.

The article has been checked for placing the references at the end of sentences, and the incorrectly placed reference in line 43 of the new version has been revised.

The fragment in lines 90-91 (97-98 in the new version) refers to a meta-analysis prepared on the basis of a multicenter European study on the epidemiology of achondroplasia; therefore, only one reference is cited and the sentence has been corrected to be more understandable.

In lines 91-94 (100-103 in the new version), the authors refer to a European meta-analysis in which, out of 434 patients, only 208 had a known medical family history. Out of those 208 patients, de novo mutation was detected in 166 cases (79.8%), while familial disease was confirmed in 42 cases (20.2%). The paragraph has been corrected to make it easier for readers to understand.

In line 108 (118 in the new version), the truncated form has been changed to the full form.

In line 111 (121 in the new version), the word 'transformation' has been changed into 'mutation'.

We have decided not to add a figure on the FGFR3 receptor to avoid plagiarism. Readers can review the physiological function of FGFR3 and its gain-of-function mutation in the pathogenesis paragraph and references 15, 17, 18.

Due to the mutually exclusive message of the available publications, we have decided to remove the fragment in lines xxx.

In lines 366-384, we described TransCon CNP study, so we used only one reference. Both mouse and monkey studies were performed in the TransCon CNP study, and we have supplemented the information on the results obtained in mouse models (lines 398-403 in the new version).

The study with Infigratinib in the context of the treatment of achondroplasia is described (there is only one). The remaining studies concern the use of this drug in different types of cancer, so we have changed the word 'trials' in the line 462 (483 in the new version) into the word 'trial'.

In order to make it easier for the reader to use our review and following your suggestions, we have decided to divide the drugs in Tables 2 and 3 into those used clinically and those that are in different phases of clinical trials.

The article has been revised for the correct use of abbreviations; 'fibroblast growth receptor 3' in line 236 (248 in the new version) has been changed into FGFR3, and the abbreviation CNP in line 299 (310 in the new version) has been defined.

The review has been standardized in terms of using decimal point in lines 98, 102, 182, 367-369, 421 of the new version.

The English language has been carefully revised and has been send to English pre-edit services of MDPI.

Round 2

Reviewer 1 Report

The manuscript has been improved therefore it is now suitable for publication

Reviewer 2 Report

The manuscript improved after the revision. I only noticed that table 3 is not cited in the text, the authors should check the text and cite the appropriate tables.